# Prenatal light exposure affects number sense and the mental number line in young domestic chicks

**Rosa Rugani[1]\*, Matteo Macchinizzi[1], Yujia Zhang[2], Lucia Regolin[1]**

[1]Department of General Psychology, University of Padua, Padova, Italy; [2]Department of Psychology, The Ohio State University, Columbus, United States

## eLife Assessment

This **fundamental** study demonstrates how a left-right bias in the relationship between numerical magnitude and space depends on brain lateralization. The evidence is **compelling** and will be of interest to researchers studying numerical cognition, brain lateralization, and cognitive brain development more broadly.

**\*For correspondence:**
rosa.rugani@unipd.it

**Competing interest:** The authors declare that no competing interests exist.

**Abstract** Humans order numerosity along a left-to-right mental number line (MNL), traditionally considered culturally rooted. Yet, some species at birth show spatial-numerical associations (SNA), suggesting neural origins. Various accounts link SNA to brain lateralization but lack evidence. We investigated brain lateralization effects on numerical spatialization in 100 newborn domestic chicks. In ovo light exposure yielded strongly lateralized brains in half the chicks and weakly lateralized in the other half. Chicks learned to select the 4th item in a sagittal array. At the test, the array was rotated 90°, with left and right 4th items correct. Strongly lateralized chicks outperformed weakly lateralized ones when ordinal and spatial cues were reliable (experiment 1), but not with unreliable spatial cues (experiment 2). Moreover, only strongly lateralized chicks showed left-to-right directionality, suggesting the right hemisphere's key role in integrating spatial and numerical cues. We demonstrate that brain lateralization is fundamental for developing a left-to-right oriented SNA.

## Introduction

Humans share with other animals basic numerical capacities (*Cantlon and Brannon, 2007*; *Cordes et al., 2001*; *Cordes and Brannon, 2009*; *Rugani, 2018*; *Vallortigara, 2018*); this evolutionarily ancient number sense serves as a building block for our unique mathematical abilities (*Dehaene, 2011*). A distinctive feature of the human numerical system is the mental number line (MNL): an association of small numbers with the left and large ones with the right. Numeral visualization in various spatial configurations (right-to-left, circular, and vertical) was first reported by *Galton, 1880* (see also *Abbadie (d'), 1880*; *Seron et al., 1992*; *Tang et al., 2008*). The first experimental demonstration of left-to-right numerical spatialization was observed in a parity judgment task, where participants pressed a left or right key for odd or even numerals. Faster responses to small numbers on the left and to large numbers on the right have been described as the SNARC effect (spatial-numerical association of response codes): a spatial congruency between the response side (left or right) and the relative position of the represented numerical magnitude on an oriented MNL (left space/small numbers and right space/large numbers) (*Dehaene et al., 1993*). Left-to-right numerical spatialization has been replicated across various experiments (e.g. *Bächtold et al., 1998*; *Dehaene, 2011*); however, evidence remains inconsistent across paradigms (*Karolis et al., 2011*)

and reading direction (*Shaki et al., 2009*; *Zebian, 2005*), emphasizing the role of task-dependent factors and culture in shaping numerical spatialization. Moreover, spatial biases in numerical processing that demonstrated spatial-numerical association (SNA) have been reported in young infants (*Bulf et al., 2016*; *de Hevia, 2021*; *de Hevia and Spelke, 2010*; *Rugani et al., 2022*; *West and McCrink, 2021*), newborns (*de Hevia et al., 2014*; *de Hevia et al., 2017*; *Di Giorgio et al., 2019*; *McCrink et al., 2020*), and nonhumans (*Drucker and Brannon, 2014*; *Giurfa et al., 2022*; *Rugani et al., 2015a*; *Rugani et al., 2010*; *Rugani et al., 2024*; *Rugani et al., 2007*; *Rugani et al., 2020b*), challenging the predominant role of culture in determining the left-to-right orientation. Evidence in nonhuman animals indicates that nervous systems across various species, despite their different levels of complexity, are prewired in how they relate numbers to space. Together, conflicting evidence of an oriented representation of numerical magnitude, as described in the MNL in humans, and broader spatial biases in numerical processing, as reported in the SNA in nonverbal or preverbal subjects, suggests that numerical spatialization likely originates from neural bases, yet it remains flexible and subject to modulation by experience and contextual factors (*Rugani and de Hevia, 2017*).

The first evidence of numerical spatialization in nonhuman subjects was observed in day-old domestic chicks (*Rugani et al., 2007*; *Rugani et al., 2010*) and adult Clark's nutcrackers (*Rugani et al., 2010*). Birds learned to identify a target item, e.g., the 4th, in a sagittally oriented array of identical items. They were then tested with the array rotated by 90°, thus laying on a fronto-parallel plane with the items oriented left to right. Animals selected the 4th left item more often than the 4th right one, suggesting that number is intrinsically represented from left to right (*Rugani et al., 2007*). Using this paradigm, the same left-to-right proto-counting tendency has also been observed in adult rhesus monkeys (*Drucker and Brannon, 2014*) and preschoolers (*West and McCrink, 2021*).

Behavioral observations supporting the SNA have increased (*Adachi, 2014*; *Bulf et al., 2016*; *de Hevia et al., 2017*; *de Hevia and Spelke, 2010*; *Di Giorgio et al., 2019*; *Drucker and Brannon, 2014*; *Giurfa et al., 2022*; *McCrink et al., 2020*; *Rugani et al., 2024*; *Rugani et al., 2010*; *West and McCrink, 2021*; but see for null results; *Beran et al., 2019*; *Triki and Bshary, 2018*). Concurrently, specific neurons responding to numerosities have been identified in human (*Pearson et al., 2009*; *Piazza et al., 2004*; *Piazza and Eger, 2016*), monkey (*Nieder et al., 2002*), crow (*Ditz and Nieder, 2015*), and chick (*Kobylkov et al., 2022*) brains. Nevertheless, the biological mechanisms and neural underpinnings of SNA are still largely unknown.

The right hemisphere is specialized for analyzing spatial information (*Rashid and Andrew, 1989*; *Regolin et al., 2005*) and for processing numerical information (*Piazza and Eger, 2016*; *Rugani and Regolin, 2020*). This functional overlap (*Harvey et al., 2013*; *Zorzi et al., 2002*) has been proposed as the basis for the SNA that prompts animals to start counting from left to right. This will be referred to as the right-hemisphere dominance model (*Rugani et al., 2016*). Two other explanatory models have been put forward: the emotional valence (*Vallortigara, 2018*) and the brain asymmetric frequency tuning (BAFT) (*Felisatti et al., 2020*).

The emotional valence model is grounded in evidence demonstrating that the right hemisphere processes negative emotions and the left hemisphere handles positive ones (*Davidson, 2004*; *Vallortigara, 2018*). This model posits that small numerosities are associated with a negative valence and activate the right hemisphere that directs movement toward the left; large numerosities would be associated with a positive valence, activating the left hemisphere that leads movement toward the right (*Vallortigara, 2018*).

BAFT links spatial frequency processing to higher cognitive functions. The right hemisphere specializes in processing low spatial frequencies, while the left hemisphere, high spatial frequencies (*Gazzaniga, 2000*). Because smaller numerosities correspond to low frequencies, and large numerosities to high frequencies, such hemispheric specialization may explain SNA (*Felisatti et al., 2020*).

Although all models, whether mutually exclusive or not, acknowledge the role of hemispheric specialization in SNA, no research has directly investigated its effect on number spatialization. Direct manipulation of brain lateralization is essential for determining its impact on SNA. While a stronger bias is expected in highly lateralized chicks, the right-hemisphere model predicts a left bias, the emotional valence model predicts a right bias, and BAFT predicts no bias.

The aim of the present study is to assess whether prenatal light stimulation influences SNA. Specifically, the study has two main objectives: (i) to assess the effect of hemispheric lateralization on number

spatialization by varying its degree; (ii) to disentangle the role of the two cerebral hemispheres using monocular occlusion.

As for the first aim, lateralization levels in domestic chicks can be easily manipulated by controlling exposure to light in the final period of incubation, between embryonic days 18 and 21 (*Rogers and Bolden, 1991*; *Rogers and Sink, 1988*). During development, the embryo rotates such that the right eye faces outward toward the translucent eggshell and any available environmental light. In contrast, the left eye is oriented toward the body, receiving little to no light. The chick embryo's visual system, specifically the thalamofugal pathway, undergoes differentiation in ovo. Egg exposure to light leads to an asymmetrical stimulation of the two eyes such that there is an increase in forebrain projections from the left side of the thalamus (fed by the light-stimulated right eye) compared with the right side (*Deng and Rogers, 1998*). Such asymmetries extend to the strength of visual projections from the thalamus to the visual Wulst (a laminated bulge in the dorsal telencephalon, functionally analogous to mammalian visual cortex; *Clark and Colombo, 2020*; *Rogers and Sink, 1988*). Specifically, the right visual Wulst receives more bilateral information from the two eyes than the left one (*Costa-lunga et al., 2022*). Evidence shows that as little as 2 hr light exposure prior to hatching is sufficient to induce these brain asymmetries (*Deng and Rogers, 1997*; *Rogers, 1982*). As a consequence of light-enhanced hemispheric differentiation, light-incubated chicks, Li-chicks, are strongly lateralized, as demonstrated by cognitive and behavioral biases (*Daisley et al., 2009*). For the present study, we expect that Li-chicks will exhibit a clearer left-to-right oriented numerical spatialization compared to dark-incubated chicks, Di-chicks, whose brain lateralization is mainly prevented (*Rogers and Bolden, 1991*).

Additionally, higher levels of lateralization enhance cognitive performance (*Daisley et al., 2009*). Strongly lateralized chicks perform better in dual tasks requiring simultaneous foraging and predator monitoring (*Dharmaretnam and Rogers, 2005*), in discrimination and categorization tasks such as distinguishing pebbles from food grains (*Deng and Rogers, 1997*), and in transitive inference tasks where they learn hierarchical stimulus relationships (A>B>C>D>E) that then apply to novel pairings (i.e. AE and BD) (*Daisley et al., 2010*). Brain lateralization also seems to improve numerical ability: strongly lateralized fish outperform non-lateralized individuals in quantity discrimination tasks. In a shoal-choice task, non-lateralized fish failed to distinguish between four and six conspecifics, while lateralized fish reliably selected the larger group. Similarly, in an abstract dot discrimination task, lateralized fish successfully distinguished between three and four dots, whereas non-lateralized fish did not (*Dadda et al., 2015*). Although lateralization was inferred indirectly through the mirror test, similar results in shoal-choice tasks emerged using a detour test (*Gatto et al., 2019*), supporting the hypothesis that brain lateralization enhances cognitive efficiency and was a key selective force in evolution (*Dadda et al., 2015*). In this study, rather than inferring brain lateralization from behavioral tests, we directly examined the impact of prenatal light stimulation, which is known to influence lateralization in domestic chickens (*Costalunga et al., 2022*; *Rogers, 1982*; *Rogers and Bolden, 1991*). This approach provides a more structural understanding of its potential effects on numerical abilities and the relationship between environmental factors and number sense. Based on previous findings, we hypothesized that numerical performance would be enhanced in Li-chicks compared to Di-chicks.

As for the second aim, to investigate hemispheric dominance, we used temporary monocular occlusion by applying a removable eye patch to cover one eye of the chicks. Due to the complete decussation of fibers at the optic chiasm (*Weidner et al., 1985*) and lack of a structure homologous to the corpus callosum (even though other smaller tracts allow interhemispheric communication; *Robert and Cuénod, 1969*; *Theiss et al., 2003*), information received by each eye is mainly elaborated by the contralateral hemisphere (*Deng and Rogers, 1998*; *Rogers and Vallortigara, 2021*). Restriction of the visual input by monocular occlusion will allow us to disentangle how the two hemispheres elaborate ordinal information and to test whether one hemisphere is dominant.

## Results

Here, we tested 100 male domestic chicks (*Gallus gallus*) of the Aviagen ROSS 308 line (experiment 1, *n*=48; experiment 2, *n*=52). Dark-incubated chicks (Di-chicks; *n*=24 in experiment 1, *n*=26 in experiment 2) were obtained from eggs incubated in darkness, while light-incubated chicks (Li-chicks; *n*=24 in experiment 1, *n*=26 in experiment 2) were obtained from eggs exposed to light (LED 4.8 W light-bulb) from days 18 to 21 of incubation.

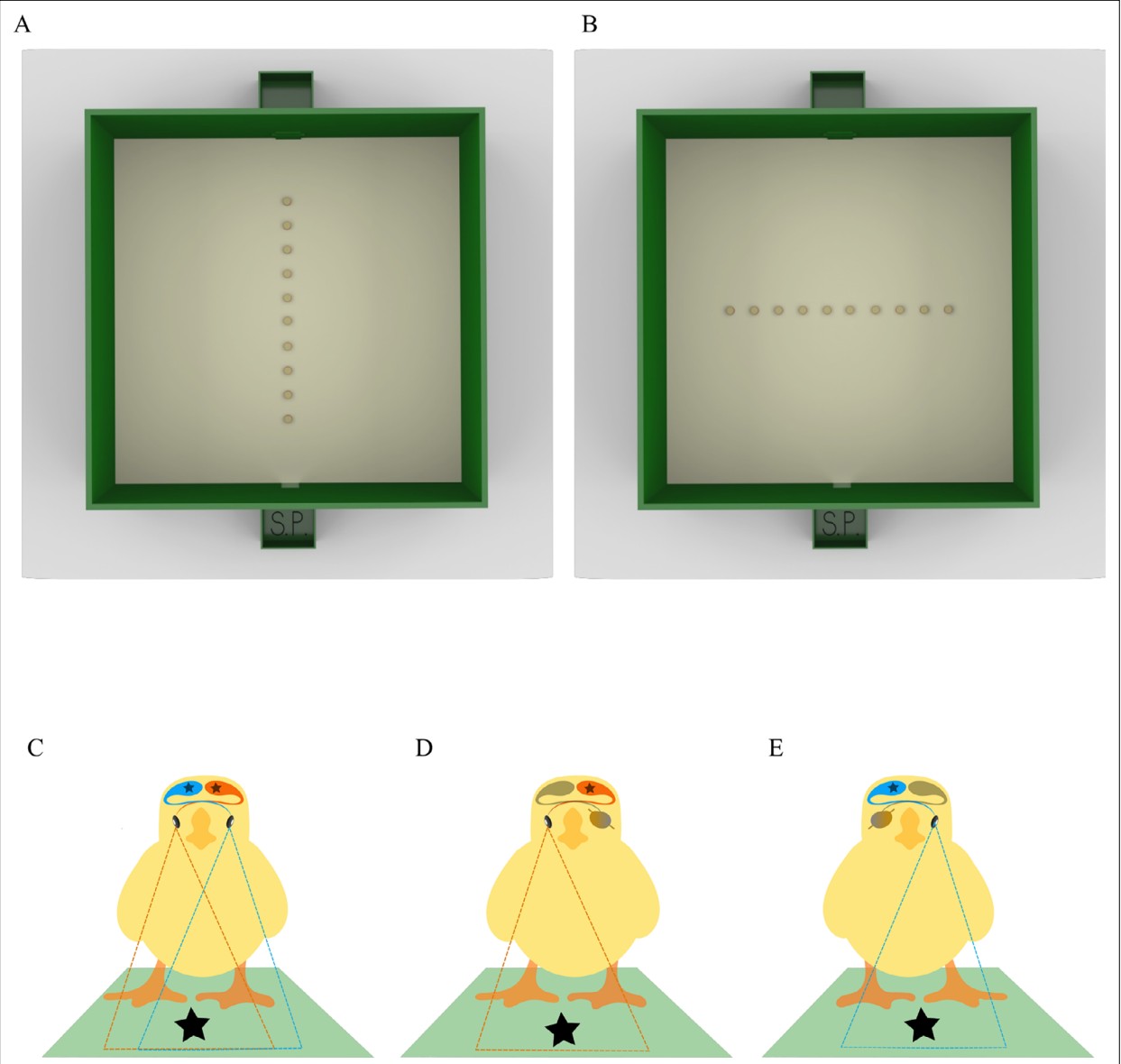

**Figure 1.** Illustration of the experimental apparatus and of the three conditions of vision. (**A**) Array configuration used at training and sagittal test. (**B**) Array configuration used for the fronto-parallel tests. S.P. indicates the chick's starting position. (**C**) Binocular condition of vision: both eyes, thus both hemispheres (left in red and right in blue), in use. (**D**) Right monocular condition with the right eye and the left hemisphere in use, in red. (**E**) Left monocular condition with the left eye and the right hemisphere in use, in blue. In monocular conditions, the eye not receiving light and its contralateral hemisphere are depicted in gray.

All chicks were trained to peck for food reward at the 4th item in an array of 10 identical, equally spaced, and sagittally aligned items (plastic caps; *Figure 1A*). Each chick underwent four tests. In the first test, the array was oriented as during training, i.e., sagittally with respect to the chick's starting point (sagittal test; *Figure 1A*). This test was run in binocular condition of vision.

Subsequently, each chick underwent tests in which the array was rotated by 90° so as to lay fronto-parallel with respect to the chick's starting position (fronto-parallel tests; *Figure 1B*). Fronto-parallel tests were administered to the chick in three conditions of vision: binocular (*Figure 1C*), monocular right (left eye patched and right eye in use; *Figure 1D*), and monocular left (right eye patched and left eye in use; *Figure 1E*).

The fronto-parallel test presented the array as oriented from left to right. This orientation produces two possible correct options, both equidistant from the subject: the 4th left and the 4th right item (*Figure 1B*). The critical difference between experiments 1 and 2 was that, during testing, spatial

**Table 1.** Models and hypotheses explaining spatial-numerical association (SNA).

For each of the three main models, a brief explanation of the hypotheses regarding SNA mechanisms and the predicted outcomes for experiments 1 and 2 is reported.

| Model | Hypothesis | Predictions for experiment 1 | Predictions for experiment 2 |
|---|---|---|---|
| Right hemisphere dominance | Right hemisphere dominance in visuospatial attention. | Right hemisphere dominance leads to left-to-right scanning. The left bias should be more pronounced in light-incubated chicks. | Absence of spatial cues results in diminished involvement of the right hemisphere, and the left bias is predicted to be less pronounced. |
| Emotional valence | Left hemisphere dominance in processing positive emotions; right hemisphere dominance in processing negative emotions. | Left hemisphere processing of positive rewards (e.g. food) leads to a rightward bias. This should be more pronounced in light-incubated chicks. | Lack of spatial cues is not expected to cause differences compared to the predictions of Experiment 1. |
| Brain asymmetric frequency tuning (BAFT) | Left hemisphere dominance in high-frequency processing; right hemisphere dominance in low-frequency processing. | No bias is expected due to the symmetrical array configuration. No difference is expected between light- and dark-incubated chicks. | Lack of spatial cues is expected to have no significant impact on the predicted outcomes of experiment 1. |

information was available in experiment 1 but was unreliable in experiment 2. Specifically, in experiment 1, the inter-item distance and the total length of the array were kept constant and identical to the training array; thus, chicks could identify the items on the basis of either ordinal or spatial cues. In experiment 2, the inter-item distance was uniform within each trial but varied systematically between trials (1.44 cm, 2.55 cm, 3.11 cm, and 3.66 cm resulting in total array lengths of respectively 43.0, 53.0, 58.0, and 63.0 cm); thus, the spatial information was unreliable.

These experimental manipulations allow us to test and compare the three models proposed to explain the origin of the MNL (*Table 1*).

1. According to the right hemisphere dominance model (*Rugani et al., 2016*), which highlights the role of the right hemisphere in processing spatial and numerical cues, a left bias is expected whenever both eyes and hemispheres are in use, particularly in individuals with greater interhemispheric differences, i.e., Li-chicks. Moreover, the left bias should be more pronounced when only the right hemisphere processes information, as in the monocular left eye condition.
2. Following the emotional valence model (*Vallortigara, 2018*), which suggests that the left hemisphere processes positive emotions, considering that food itself is associated with positive emotions, a right bias might be expected in bilateral processing during food search. This bias should be more pronounced in Li-chicks due to their enhanced hemispheric differentiation.
3. As for the BAFT model (*Felisatti et al., 2020*), the left/right symmetry in item disposition within the array would lead to an absence of bias with no difference between strong and weak lateralized animals.

In each trial, chicks were allowed a single peck. We recorded the selected item to calculate the percentage of responses at each position and averaged them separately for each group and test. We analyzed the group percentage for choosing the 4th item above chance (10%), using Bonferroni correction for multiple comparisons (data and significant results are reported in *Table 2*; additional analyses on the selection of each item are reported in *Supplementary file 1*; *Supplementary file 2*). To assess side bias in the fronto-parallel tests, we compared correct choices on the left (4L) vs. the right (4R) using a paired *t*-test, with Cohen's *d* as the effect size, and Bonferroni correction (see *Table 2*). Moreover, we tested whether brain lateralization influenced accuracy by comparing the percentage of correct choices (i.e. the selection of the 4th item in the sagittal test; the 4L or 4R items in the fronto-parallel tests) between Li-chicks and Di-chicks using a two-sample *t*-test, with Cohen's *d* as the effect size, and Bonferroni correction. We conducted both frequentist and Bayesian statistics to ensure reliable interpretations of our results.

## Results in the tests allowing utilization of reliable ordinal and spatial cues (experiment 1)

### Sagittal test conducted under binocular vision condition

In the sagittal test (*Figure 2A*), Li-chicks selected the 2nd and 4th items above chance; Di-chicks selected the 1st, 2nd, and 4th items above chance (*Table 2* and *Supplementary file 1*). Moreover,

**Table 2.** Descriptive statistics.

For each test in the two experiments, the accuracy of selecting the 4th item in the sagittal test and the 4L or 4R items in the fronto-parallel (FP) tests is reported.

Experiment 1: Ordinal and spatial cue

| Test | Hatch condition | Choice | mean | SD | SE | n | r | p | BF |
|---|---|---|---|---|---|---|---|---|---|
| Sagittal | Di-chicks | 4 | 28.436 | 9.374 | 1.913 | 24 | 0.878 | <0.001 | >10,000 |
| | Li-chicks | 4 | 36.382 | 8.164 | 1.666 | 24 | 0.88 | <0.001 | >10,000 |
| FP binocular | Di-chicks | 4L | 18.958 | 10.527 | 2.149 | 24 | 0.727 | 0.005 | 169 |
| | | 4R | 19.792 | 11.371 | 2.321 | 24 | 0.724 | 0.003 | 188 |
| | Li-chicks | 4L | 28.851 | 12.598 | 2.572 | 24 | 0.859 | <0.001 | >10,000 |
| | | 4R | 11.894 | 7.503 | 1.532 | 24 | 0.337 | 0.798 | 0.741 |
| FP monocular left | Di-chicks | 4L | 12.780 | 8.309 | 1.696 | 24 | 0.376 | 0.532 | 1.287 |
| | | 4R | 3.990 | 6.154 | 1.256 | 24 | –0.708 | 1.000 | 0.05 |
| | Li-chicks | 4L | 22.595 | 8.736 | 1.783 | 24 | 0.876 | <0.001 | >10,000 |
| | | 4R | 6.297 | 6.747 | 1.377 | 24 | –0.465 | 1.000 | 0.068 |
| FP monocular right | Di-chicks | 4L | 3.746 | 3.668 | 0.749 | 24 | –0.798 | 1.000 | 0.013 |
| | | 4R | 13.417 | 8.781 | 1.792 | 24 | 0.406 | 0.398 | 1.936 |
| | Li-chicks | 4L | 5.219 | 6.356 | 1.297 | 24 | –0.594 | 1.000 | 0.057 |
| | | 4R | 21.484 | 10.734 | 2.191 | 24 | 0.807 | 0.001 | 1843 |

Experiment 2: Ordinal cue only

| Test | Hatch condition | Choice | mean | SD | SE | n | r | p | BF |
|---|---|---|---|---|---|---|---|---|---|
| Sagittal | Di-chicks | 4 | 31.487 | 15.337 | 3.008 | 26 | 0.863 | <0.001 | >10,000 |
| | Li-chicks | 4 | 32.275 | 11.232 | 2.203 | 26 | 0.876 | <0.001 | >10,000 |
| FP binocular | Di-chicks | 4L | 18.725 | 9.246 | 1.813 | 26 | 0.74 | 0.001 | 836.246 |
| | | 4R | 19.666 | 10.024 | 1.966 | 26 | 0.786 | 0.002 | 1070.73 |
| | Li-chicks | 4L | 21.771 | 9.888 | 1.939 | 26 | 0.819 | <0.001 | >10,000 |
| | | 4R | 15.800 | 9.239 | 1.812 | 26 | 0.603 | 0.024 | 21.919 |
| FP monocular left | Di-chicks | 4L | 14.835 | 9.878 | 1.937 | 26 | 0.473 | 0.122 | 5.337 |
| | | 4R | 8.333 | 6.912 | 1.356 | 26 | –0.316 | 1.000 | 0.102 |
| | Li-chicks | 4L | 15.011 | 9.344 | 1.832 | 26 | 0.518 | 0.050 | 8.449 |
| | | 4R | 7.939 | 6.082 | 1.193 | 26 | –0.302 | 1.000 | 0.084 |
| FP monocular right | Di-chicks | 4L | 8.290 | 7.659 | 1.502 | 26 | –0.269 | 1.000 | 0.102 |
| | | 4R | 12.457 | 9.255 | 1.815 | 26 | 0.29 | 0.850 | 0.839 |
| | Li-chicks | 4L | 6.288 | 6.151 | 1.206 | 26 | –0.515 | 1.000 | 0.06 |
| | | 4R | 17.984 | 9.634 | 1.889 | 26 | 0.78 | 0.001 | 214 |

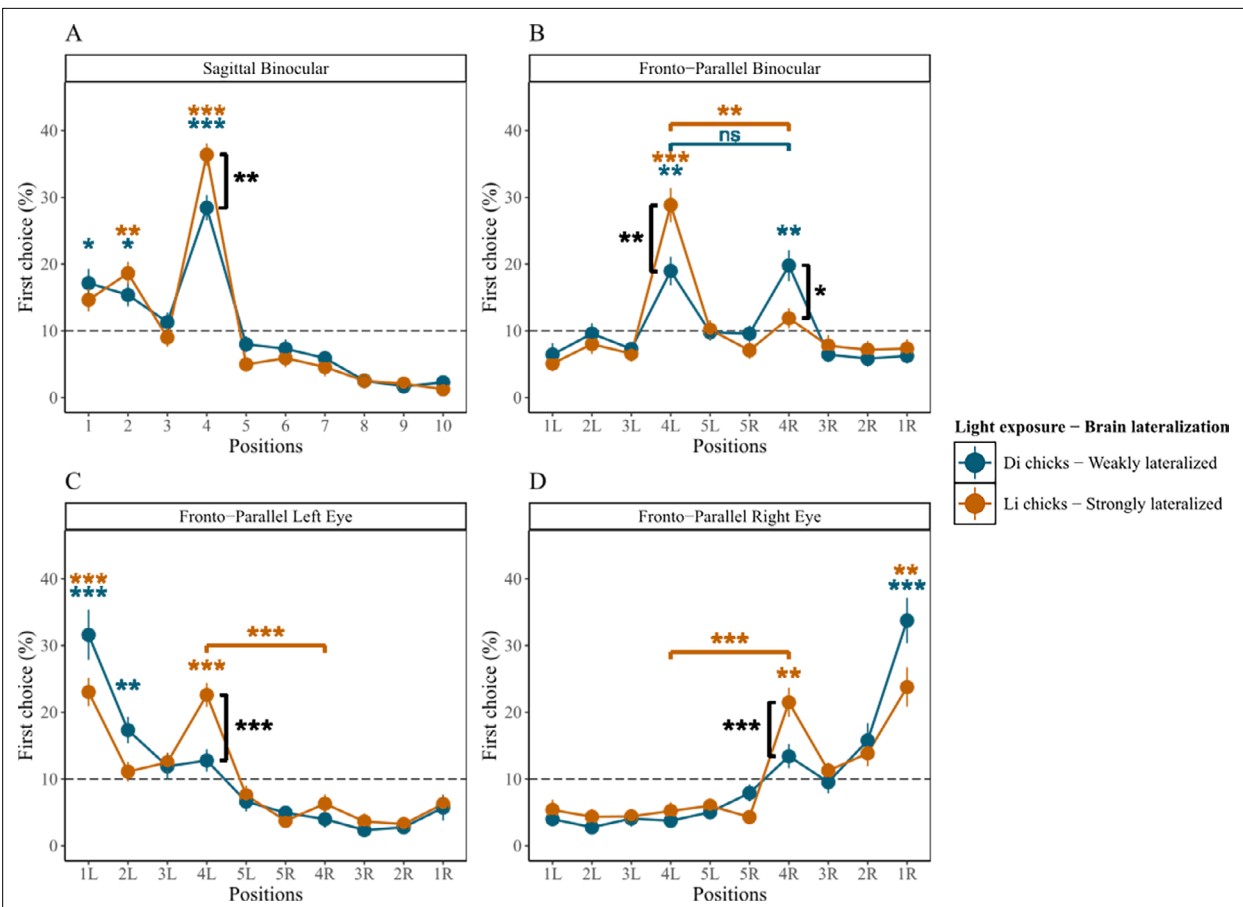

**Figure 2.** Results of Experiment 1. The average percentage of chicks' choices (y-axis) as a function of item positions (x-axis), light exposure modulating brain lateralization, and tests. Error bars indicate ± standard error. Di chicks: n = 24, Li chicks: n = 24 (consistent across all tests). The gray dashed line indicates chance level (10%). Significant deviations from chance level were assessed through Wilcoxon one-sample signed-rank tests with Bonferroni correction (P < 0.001, P < 0.01, P < 0.05). (**A**) Results of the Sagittal test. (**B**) Results of the Fronto-Parallel Binocular test. (**C**) Results of the Fronto-Parallel Monocular Left test. (**D**) Results of the Fronto-Parallel Monocular Right test.

Li-chicks showed higher accuracy in selecting the 4th item than Di-chicks (Li-chicks: $n$=24, mean = 36.38, SE = 1.666; Di-chicks: $n$=24, mean = 28.44, SE = 1.913; $t$(45.1)=3.132, p=0.003, $d$=0.904; BF = 12.588) (**Figure 2A**).

### Fronto-parallel transfer test conducted under binocular vision condition

In the fronto-parallel binocular test (**Figure 2B**), Li-chicks successfully transferred what was learned to the rotated series: they selected only the 4th left item above chance (**Table 2**). Li-chicks showed a left bias, pecking the 4th left item more than the 4th right one ($n$=24, $t$(23) = 4.791, p<0.001, $d$=1.635; BF = 337.124). Di-chicks pecked both 4th left and 4th right items above chance (**Table 2**) without a difference ($n$=24, $t$(23) = 0.218, p=1.000 $d$=0.076; BF = 0.219). Remarkably, only Li-chicks showed a left bias, indicating a tendency to proto-count from left to right.

Comparing accuracy between the two groups, Li-chicks selected the 4th left item more than Di-chicks did (Li-chicks: $n$=24, mean = 28.85, SE = 2.572; Di-chicks: $n$=24, mean = 18.96, SE = 2.149; $t$(44.6)=2.952, p=0.010, $d$=0.852; BF = 8.456), while Li-chicks chose the 4th right item less than Di-chicks (Li-chicks: $n$=24, mean = 11.89, SE = 1.532; Di-chicks: $n$=24, mean = 19.79, SE = 2.321; $t$(39.8)=–2.840, p=0.014, $d$=0.820; BF = 6.657).

### Fronto-parallel transfer test conducted under left monocular vision condition

In the fronto-parallel monocular left test (**Figure 2C**), Li-chicks were able to transfer learning to a differently oriented series, correctly selecting the 4th left item above chance, even if they also pecked

at the 1st left item (*Table 2*; *Supplementary file 1*). Moreover, they selected the 4th left more than the 4th right item ($n=24$, $t(23) = 6.056$, p<0.001, $d=2.088$; BF = 5598.452). Di-chicks failed: they only selected the 1st and 2nd left items above chance (*Supplementary file 1*).

As for the difference between the two groups, Li-chicks selected the 4th left item more than Di-chicks (Li-chicks, $n=24$, mean = 22.59, SE = 1.783; Di-chicks, $n=24$, mean = 12.78, SE = 1.696; $t(45.9)=3.988$, p<0.001, $d=1.151$; BF = 104.830).

These results suggest that whenever the right hemisphere is processing the information, light exposure affects the left bias and numerical performance. This evidence, on one side, supports the relevance of the right hemisphere in directing SNA directionality (*Rugani et al., 2015a*; *Rugani and de Hevia, 2017*). On the other side, it shows how significant experiences that stimulate brain development, although limited to a few hours of exposure to moderate ambient light, can boost cognitive performance.

## Fronto-parallel transfer test conducted under right monocular vision condition

In the fronto-parallel monocular right test (*Figure 2D*), Li-chicks succeeded correctly in selecting the 4th right item, which was pecked more than the 4th left item ($n=24$, $t(23) = -6.151$, p<0.001, $d=1.844$; BF = 6887.511), even if they also selected the 1st left item above chance (*Table 2* and *Supplementary file 1*). Di-chicks failed: they did not select the 4th left nor right item above chance; the only item selected above chance was the 1st right one (*Supplementary file 1*).

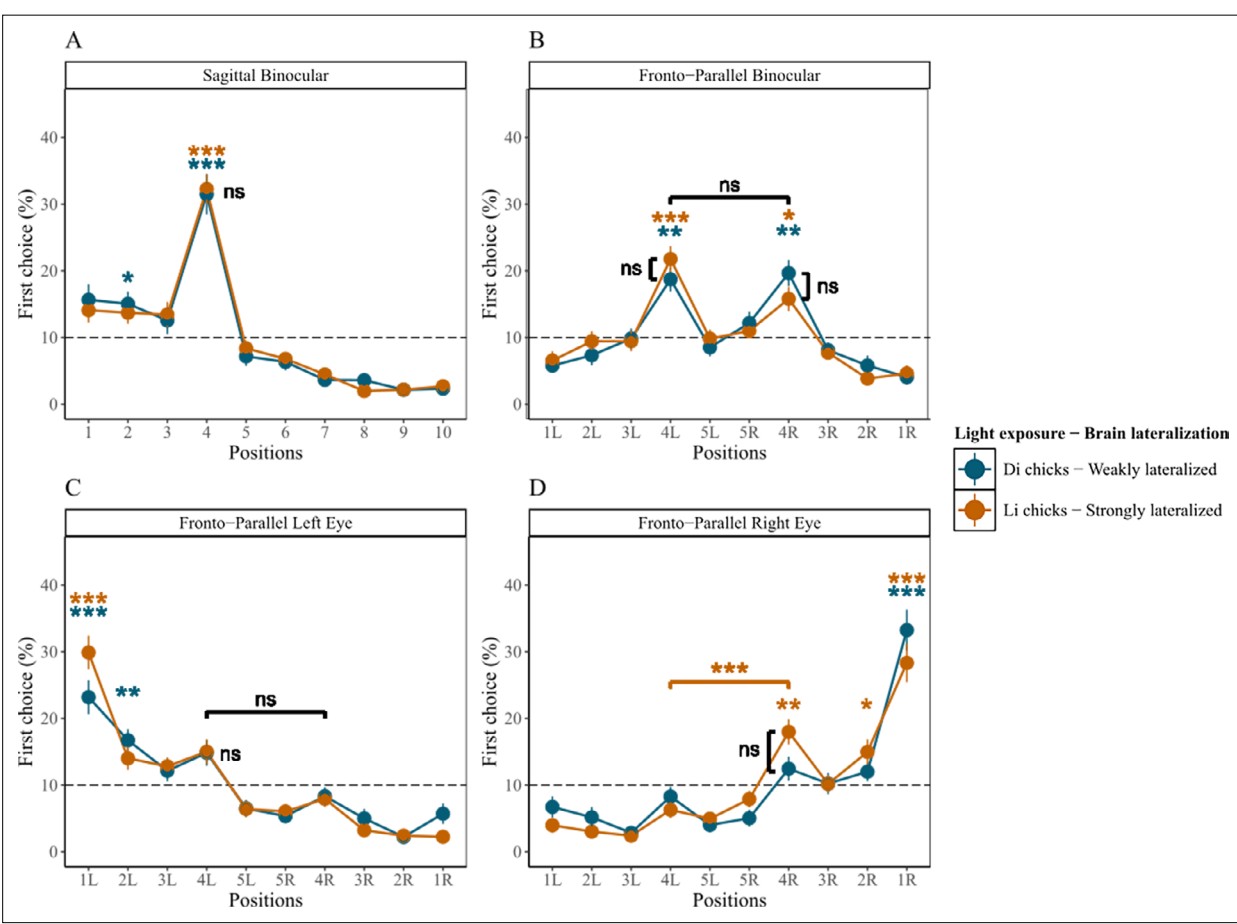

**Figure 3.** Results of Experiment 2. The average percentage of chicks' choices (y-axis) as a function of item positions (x-axis), light exposure modulating brain lateralization, and tests. Error bars indicate ± standard error. Di chicks: n = 26, Li chicks: n = 26 (consistent across all tests). The gray dashed line indicates chance level (10%). Significant deviations from chance level were assessed through Wilcoxon one-sample signed-rank tests with Bonferroni correction (P < 0.001, P < 0.01, P < 0.05). (**A**) Results of the Sagittal test. (**B**) Results of the Fronto-Parallel Binocular test. (**C**) Results of the Fronto-Parallel Monocular Left test. (**D**) Results of the Fronto-Parallel Monocular Right test.

As for the accuracy differences between the two groups, Li-chicks selected the 4th right item more than Di-chicks (Li-chicks: $n$=24, mean = 21.48, SE = 2.191; Di-chicks: $n$=24, mean = 13.42, SE = 1.792; $t$(44.3)=2.850, p=0.013, $d$=0.823; BF = 6.799).

Again, only Li-chicks succeeded while Di-chicks failed, corroborating evidence on the importance of light stimulation in favoring the development of both hemispheres (*Costalunga et al., 2022*) and boosting cognitive performance.

Overall, these data showed that prenatal light experience can stimulate brain development and hemispheric specialization, which emphasizes the SNA and enhances performance in a spatial/numerical task. This provides novel evidence of the role of brain lateralization in determining SNA and in boosting proto-numerical counting.

## Results in the tests allowing utilization of reliable ordinal and unreliable spatial cues (experiment 2)

### Sagittal test conducted under binocular vision condition

In the sagittal test (*Figure 3A*), both groups succeeded: Li-chicks exclusively selected the correct 4th item above chance (*Table 2*), while Di-chicks selected the 4th item, but also mistakenly pecked the 2nd item (*Table 2* and *Supplementary file 2*). Yet, accuracy in pecking the 4th item did not differ in the two groups (Li-chicks: $n$=26, mean = 32.27, SE = 2.203; Di-chicks: $n$=26, mean = 31.49, SE = 3.008; $t$(45.8)=0.211, p=1.000, $d$=0.059; BF = 0.283).

### Fronto-parallel transfer test conducted under binocular vision condition

In the fronto-parallel binocular test (*Figure 3B*), Li-chicks and Di-chicks selected both 4th left and right items above chance (*Table 2*). For each group, no differences emerged in selecting the 4th left and 4th right items, indicating a lack of side bias in both groups (Li-chicks: $n$=26, $t$(25) = 1.910, p=0.135, $d$=0.624; BF = 0.997; Di-chicks: $n$=26, $t$(25) = –0.218, p=1.000, $d$=0.098; BF = 0.219). Li-chicks did not differ from Di-chicks in responses to the 4th left (Li-chicks: $n$=26, mean = 21.77, SE = 1.939; Di-chicks: $n$=26, mean = 18.73, SE = 1.813; $t$(49.8)=1.147, p=0.513, $d$=0.318; BF = 0.478) or 4th right item (Li-chicks: $n$=26, mean = 15.80, SE = 1.812; Di-chicks: $n$=26, mean = 19.67, SE = 1.966; $t$(49.7)=–1.446, p=0.309, $d$=0.401; BF = 0.655). This confirms that whenever the spatial information is unavailable at the test, the left bias disappears (*Rugani et al., 2011*), highlighting the role of the right hemisphere in processing spatial information and determining the left-to-right orientation of the SNA.

### Fronto-parallel transfer test conducted under left monocular vision condition

In the fronto-parallel left monocular test (*Figure 3C*), Li-and Di-chicks failed: Li-chicks selected the 1st left item (*Supplementary file 2*); Di-chicks selected the 1st and the 2nd left item above chance (*Supplementary file 2*). As for the accuracy, the two groups equally selected the 4th left (Li-chicks: $n$=26, mean = 15.01, SE = 1.832; Di-chicks: $n$=26, mean = 14.84, SE = 1.937; $t$(49.8)=0.066, p=1.000, $d$=0.018; BF = 0.279) and the 4th right items (Li-chicks: $n$=26, mean = 7.94, SE = 1.193; Di-chicks: $n$=26, mean = 8.33, SE = 1.356; $t$(49.2)=–0.218, p=1.000, $d$=0.060; BF = 0.284).

Whenever spatial information is unavailable, the right hemisphere fails to transfer tasks. This indicates that independent of the hemisphere's development, unilateral right hemispheric processing is insufficient in dealing with an ordinal task; thus, ordinality does not appear to be lateralized to the right hemisphere.

### Fronto-parallel transfer test conducted under right monocular vision condition

In the fronto-parallel monocular right test (*Figure 3D*), only Li-chicks succeeded and selected the 4th right item; even if they also pecked the 1st and the 2nd right items (*Table 2* and *Supplementary file 2*), moreover they selected the 4th right more than the 4th left item ($n$=26, $t$(25) = –4.946, p<0.001, $d$ = –1.447; BF = 573.520). Di-chicks failed to select the 4th left or right item; instead, they selected only the 1st right item above chance (*Supplementary file 2*). As for the difference in accuracy between the two groups, Li-chicks did not select the 4th right item more than Di-chicks (Li-chicks: $n$=26, mean = 17.98, SE = 1.889; Di-chicks: $n$=26, mean = 12.46, SE = 1.815; $t$(49.9)=2.110, p=0.080, $d$=0.585; BF =

**HATCHING WITH NUMBERS**

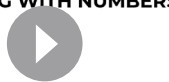

**Video 1.** Visual summary of research outcomes: illustrative video of the experimental procedure and main results.
https://elifesciences.org/articles/106356/figures#video1

1.678). This suggests that lateralization influences numerical cognition even in the absence of spatial information, and that the left hemisphere plays a significant role in processing ordinal information.

## Discussion
### General

Our main findings are that prenatal exposure that can modulate brain lateralization in domestic chicks impacts the left-to-right oriented numerical spatialization and numerical performance (*Video 1*).

In experiment 1, which allowed chicks to reliably use both ordinal and spatial cues in identifying the 4th item, chicks exhibited different behaviors despite identical learning experiences and tasks, either showing or not showing a left bias, depending on their prenatal light exposure. In the fronto-parallel binocular test, when both eyes and hemispheres processed the information, chicks hatched from light-incubated eggs, Li-chicks, selected only the 4th left item; while chicks hatched from dark-incubated eggs, Di-chicks, equally selected the 4th left and the 4th right item. Only Li-chicks (i.e. more strongly lateralized) demonstrated left-to-right proto-counting, indicating that brain lateralization influences MNL directionality. When the two hemispheres engaged in differential processing, as observed in Li-chicks, a unidirectional left-to-right oriented numerical spatialization emerged. Conversely, when prenatal stimulation did not enhance hemispheric specialization, resulting in more homogeneous hemispheric processing, animals (Di-chicks) showed no directional bias. These pioneering findings corroborate all models positing hemispheric specialization as the neural basis for SNA (*Felisatti et al., 2020*; *Rugani et al., 2016*; *Vallortigara, 2018*), while establishing lateralization as an essential prerequisite for numerical spatialization. However, pre-hatching light stimulation did not affect chicks' performance when spatial information was unavailable (experiment 2). This finding substantiates the relevance of spatial information and highlights that its integration with numerical processing within shared cortical regions is fundamental to the neurobiological underpinning of number spatialization. This integration is coherent with the fact that in the chick's brain, the right hemisphere is dominant in processing spatial information (*Rashid and Andrew, 1989*; *Regolin et al., 2005*), but can also process numerical information (*Rugani et al., 2011*; *Rugani and Regolin, 2020*) (consistently with primates and human literature; *Piazza and Eger, 2016*). The present data align with previous research that used monocular occlusion to disentangle the engagement of the two hemispheres with spatial or ordinal cues (*Rugani and Regolin, 2020*). In prior research, day-old chicks learned to select the 4th item in an array of 10 identical sagittal-aligned items maintained in fixed positions, so that both spatial and ordinal cues were available during learning. At test, chicks faced a left-to-right oriented series where the inter-item distance was manipulated so that the 3rd item was at the same distance from the beginning of the series as the 4th item had been at training. This forced chicks to choose either spatial or ordinal cues. Chicks tested binocularly selected both the 4th left and right items above chance expectation, confirming that a coherent use of numerical and spatial information is essential in limiting birds' responses toward the left (*Rugani et al., 2011*). Chicks tested monocularly chose the 3rd and 4th items on the seeing side, suggesting that birds relied on spatial or ordinal cues to a similar extent in different trials and that each hemisphere can process both cues (*Rugani and Regolin, 2020*).

Here, in monocular conditions, Li-chicks succeeded in fronto-parallel tests. Even if they directed their pecks to the visible side (*Rugani et al., 2016*; *Rugani and Regolin, 2021*), they selected the 4th item above chance expectation. This was the case in experiment 1 when ordinal and spatial cues were available at test. Nevertheless, in experiment 2, when spatial information was available at training but unreliable at test, chicks succeeded in the right eye/left hemisphere, but not in the left eye/right hemisphere condition. This corroborates the hypothesis of the right hemisphere specificity in the analysis of spatial cues and suggests that the left hemisphere is more specialized in processing ordinal information. The left-to-right spatialization of numerosity appears to be based on preferential processing

by the right hemisphere when spatial information is available and hemispheric specialization is favored by environmental stimulation.

## Ecological implications and adaptive variability

Experimental contexts allow for selective manipulation of the environment, enabling changes that are highly improbable in nature. In experiment 2, the item arrangement, that had been experienced as a stable context that provided coherent and reliable numerical and spatial information during training, was manipulated to eliminate spatial information at test. This manipulation resulted in the disappearance of left-to-right oriented directionality in the binocular condition and in a failure in the left eye/right hemisphere condition. We can speculate that in some naturalistic contexts, establishing an anchor point (potentially based on reliable landmarks or beacons) and predetermining a privileged starting position might be advantageous in facilitating processing and reducing conflicts between incompatible responses. Throughout evolution, this may have maximized right hemisphere engagement, specialized for spatial information processing, and triggered an imbalance favoring left space. This left-biased space would serve as an anchor point from which to initiate environmental scanning, avoiding the delay presumably implied if there was not a hemisphere taking control of processing and guiding behavior (*Rogers et al., 2013*). Such an intrinsic, left oriented bias might be advantageous for other ecological situations requiring number processing, such as quantifying conspecifics or food items. For example, when foraging, a consistent left-to-right scanning pattern could help animals to efficiently locate and quantify food sources without overlooking areas. The tendency to scan items from the preferred left side (*Diekamp et al., 2005*) might have evolved as an adaptive behavior to maximize fitness. This tendency could then have been assimilated by other cognitive processes that share neural substrates, including numerical cognition. Numerosities relevant for animals (such as the number of conspecifics, food items, or predators) are inherently distributed in space. The right hemisphere specialized in spatial processing incorporated some rudimentary forms of enumeration. This resulted in a right hemispheric dominance for both space and number. This might explain the observed left-to-right bias in numerical cognition tasks. The absence of this bias in conditions where spatial cues are artificially eliminated, as in experiment 2, underscores the spatial nature of numerical processing of objects in the environment. Remarkably, the left-to-right directionality is not reported in weakly lateralized Di-chicks in both experiments, irrespective of spatial cues availability. It should be noted that in some situations, like predator detection, even if a systematic approach to surveying the surroundings could be beneficial for prompt threats detection, it could also lead to more predictability. This could favor predators with a complementary approach directionality. From an evolutionary perspective, lateralization variability within a species can be viewed as an adaptive strategy. This variability may represent a form of evolutionary bet-hedging (*Simons, 2011*), where different degrees of lateralization confer different adaptive advantages in fluctuating ecological contexts. Bet-hedging strategies maintain population fitness by promoting phenotypic diversity, optimizing adaptation in unpredictable environments (*Philippi and Seger, 1989*). Lateralization variability might also contribute to the species' behavioral unpredictability, offering an advantage in predator-prey dynamics (*Güntürkün et al., 2020*). The persistence of both lateralized and non-lateralized individuals within a population may be an evolutionarily stable strategy (*Ghirlanda and Vallortigara, 2004*), conferring differential advantages to different individuals, overall maintaining high fitness and making the overall population less predictable (*Rogers, 2021*).

## Monocular test outcomes support that light-induced lateralization enhances spatial-numerical performance

Our investigation produced a second major result and showed that prenatal exposure significantly affected performance. In the sagittal test, Li-chicks outperformed Di-chicks, demonstrating that a higher degree of lateralization led to greater accuracy when both spatial and numerical cues were available (experiment 1).

The results of the monocular fronto-parallel tests further support the effect of embryonic stimulation in enhancing performance, as only Li-chicks succeeded in both the left and right monocular conditions. Remarkably, in the fronto-parallel test of experiment 1 allowing a coherent use of spatial and numerical information, Li-chicks tested in left monocular condition of vision showed a bias alike to Li-chicks tested in binocular condition of vision. When a hemisphere is dominant for a function,

behavior under its sole control often matches that observed in normal conditions of vision when both hemispheres are active (*Rogers et al., 2013*). In the present scenario, similarities between chicks tested in binocular and left monocular conditions of vision suggest that when both hemispheres are processing the information, the availability of spatial cues triggers an overactivation of the right hemisphere resulting in a leftward bias (*Rugani et al., 2016*).

Whenever processing was confined to a single hemisphere, either one, only strongly lateralized Li-chicks succeeded, while weakly lateralized Di-chicks failed, corroborating evidence on the importance of light stimulation in favoring the development and specialization of both hemispheres and in boosting cognitive performance. This finding aligns with previous anatomical studies that demonstrated the presence of light-dependent lateralization in bilaterally responsive units of the right visual Wulst (*Costalunga et al., 2022*; *Rogers and Deng, 1999*). Nevertheless, in the specific case of Li-chicks tested with the left eye/right hemisphere in use, subjects failed the fronto-parallel test when the use of spatial cues was prevented (experiment 2). This highlights the reliance of the right hemisphere on spatial information.

## Interpreting results through proposed models for the origin of the MNL

The present results reveal that brain lateralization influences performance in ordinal tasks involving both spatial and numerical cues, suggesting a joint contribution of hemispheric specialization and environmental stimulation to the spatial organization of numbers. These results allow us to reconsider the models proposed to explain the SNA.

1. The BAFT model would not predict any asymmetry as it refers number spatialization to differences in spatial frequencies (*Felisatti et al., 2020*), but in the present study, the spatial distribution of the stimuli is symmetrical. The predictions based on this model fit with the results of experiment 2, where chicks did not show any spatial bias in the fronto-parallel binocular test. Yet the model fails in predicting the left bias found in the fronto-parallel binocular and left monocular tests of experiment 1. While this model can still be valuable in explaining SNA in other contexts, it fails to account for the left-to-right proto-counting observed in this experimental setting (*Drucker and Brannon, 2014*; *Rugani et al., 2016*; *Rugani et al., 2011*; *Rugani et al., 2010*; *Rugani et al., 2007*; *West and McCrink, 2021*).

2. According to the emotional valence model, chicks should exhibit a rightward bias in food-seeking behavior, since food rewards trigger positive emotions, particularly when lateralization is pronounced. The assumption that a larger numerosity can be associated with more positive emotion and consequent preferential processing by the left hemisphere would explain the right bias (*Vallortigara, 2018*). Conversely, the assumption that a smaller numerosity is associated with a negative emotion would activate the right hemisphere, driving animals toward the left (*Vallortigara, 2018*). Nevertheless, in the present study, chicks showed a left bias when they could rely on consistent spatial and numerical information (experiment 1) and no bias when spatial information was unreliable (experiment 2). Thus, the results obtained in the present study do not fit the emotional model.

3. The left bias meets the predictions based on the right-hemisphere dominance model, which links the origin of the MNL to the right hemisphere's specialization in visual/numerical processing, resulting in a predisposed left-to-right scanning tendency (*Rugani et al., 2015a*). Remarkably, this model has been put forward to explain outcomes in both the previously mentioned paradigms (*Giurfa et al., 2022*; *Rugani et al., 2016*; *Rugani and de Hevia, 2017*), thus explaining both the left-to-right oriented searching found in ordinal tasks and the association of smaller numerosity with the left and larger numerosity with the right. Such association likely results from brain asymmetry driven by right-hemisphere dominance in visuospatial attention. Although in principle arbitrary, the left-to-right mapping direction during evolution may have been imposed by brain asymmetry: a common and ancient trait that occurs in a wide range of vertebrates (*Rogers and Andrew, 2002*; *Vallortigara and Rogers, 2005*) and invertebrates (*Anfora et al., 2011*; *Baracchi et al., 2018*; *Frasnelli et al., 2014*; *Frasnelli et al., 2012*), which possibly optimizes simultaneous processing of different kinds of information (*Rogers et al., 2013*). In natural environments, relevant numerosity (e.g. predators, food items, conspecifics) is intrinsically linked to their spatial arrangements, and their enumeration might be facilitated whenever a clear scanning directionality is present. Consequently, numerical estimation may have evolved in conjunction with spatial processing biases, leading to a left-anchored, rightward-directed environmental scanning strategy. This hypothesis suggests that numerical cognition evolved

incorporating spatial processing biases and reflecting the spatial nature of ecologically relevant numerosity.

It should be noted that BAFT and the emotional valence models have been elaborated to explain the performance in a different task (*Di Giorgio et al., 2019*; *Giurfa et al., 2022*; *Rugani et al., 2015a*). This required animals first to learn to associate a food reward with a central numerosity, e.g., an array depicting five dots. At test, when presented with new but identical numerosity, placed one on the left and one on the right, animals chose the left option when the test numerosity was smaller than the one experienced during learning, e.g., 2, and the right option when the numerosity was larger, e.g., 8 (*Di Giorgio et al., 2019*; *Giurfa et al., 2022*; *Rugani et al., 2015a*). Such a shift in the bias from left to right depending on numerosity cannot be simply associated with the lateralization of feeding responses typically guided by the left hemisphere in chicks (*Deng and Rogers, 2002*; *Deng and Rogers, 1997*; *Dharmaretnam and Rogers, 2005*). Nevertheless, numerosities could correlate with spatial frequencies (*Felisatti et al., 2020*) or with emotional valence (*Vallortigara, 2018*). Remarkably, the right hemisphere dominance model provides a more comprehensive explanation. In fact, it accounts for the above-mentioned task (*Di Giorgio et al., 2019*; *Giurfa et al., 2022*; *Rugani et al., 2015a*; *Rugani and de Hevia, 2017*), as well as for the ordinal task (*Rugani et al., 2016*; *Rugani and Regolin, 2020*).

Although all models (*Felisatti et al., 2020*; *Rugani et al., 2016*; *Vallortigara, 2018*) identify the key role of hemispheric specialization in determining numerical spatialization, the right-hemisphere dominance model can explain animal behavior across multiple contexts, providing a more parsimonious and generalizable explanation of cognitive processes, and potentially revealing fundamental connections between seemingly distinct cognitive domains.

## Materials and methods
### Subject

We tested 100 male domestic chicks (*G. gallus*) of the Aviagen ROSS 308 line (experiment 1, $n$=48; experiment 2, $n$=52). Sample size was determined by a power analysis for a multiple regression design involving four groups (test order: right-test-first vs. left-test-first×hatch condition: Di-chicks vs. Li-chicks). To detect a medium effect size ($f$=0.25) with 80% power at an alpha level of 0.05, a total sample of 48 chicks was required. Although the regression analysis is not reported in the manuscript, the full code and results are available in the article data repository. We chose male chicks because of their superior response to food reinforcement compared to females (*Regolin et al., 2005*; *Vallortigara et al., 1990*) and significantly greater degree of lateralization in the thalamofugal pathway (*Rajendra and Rogers, 1993*). The fertilized eggs were obtained weekly from two local hatcheries (Agricola Berica, Montegalda, Vicenza, Italy, or Società Agricola La Pellegrina Spa, San Pietro in Gù, Padova, Italy). Eggs on the seventh or fourteenth day of incubation were delivered to the lab and placed in a FIEM incubator MG 70/100 (45×58×43 cm$^3$) at a controlled temperature of 36–38°C and 60% humidity. On the eighteenth day of incubation, eggs were moved to a hatching machine (60×32×40 cm$^3$) at controlled temperature and humidity (36–38°C; 60%) until the 21st day of incubation (hatching day). Animals were randomly assigned to experimental conditions. Eggs were incubated under two conditions: in darkness to obtain dark-incubated (weakly lateralized) chicks (Di-chicks, $n$=24 in experiment 1, $n$=26 in experiment 2) and under light exposure using an LED 4.8 W lightbulb to obtain light-incubated (strongly lateralized) chicks (Li-chicks, $n$=24 in experiment 1, $n$=26 in experiment 2). Only the chicks that completed all four tests were included in the final sample size. A few hours after hatching, chicks were feather-sexed and caged in pairs or triplets in standard metal cages (28×32×40 cm$^3$) with the floor covered with absorbent paper. The rearing room was maintained at a temperature of 28–31°C and humidity of about 60%. The cages were illuminated by neon lights (36 W) placed about 15 cm above each cage, with a standard 24 hr light-dark rearing cycle. Food (chick crumbles) and water were available in transparent glass jars (5×5 cm$^2$) ad libitum. Daily, chicks were familiarized and fed with some mealworms (*Tenebrio molitor larvae*) that were used as reinforcement during training. These rearing conditions were maintained until the initiation of the experimental protocol on Wednesday (8 a.m.), their third day of life, when food jars were removed from cages and chicks were isolated one per cage. Chicks underwent 2 hr of food deprivation before the start of each experimental session (shaping, training, and tests). Following the last testing session, chicks

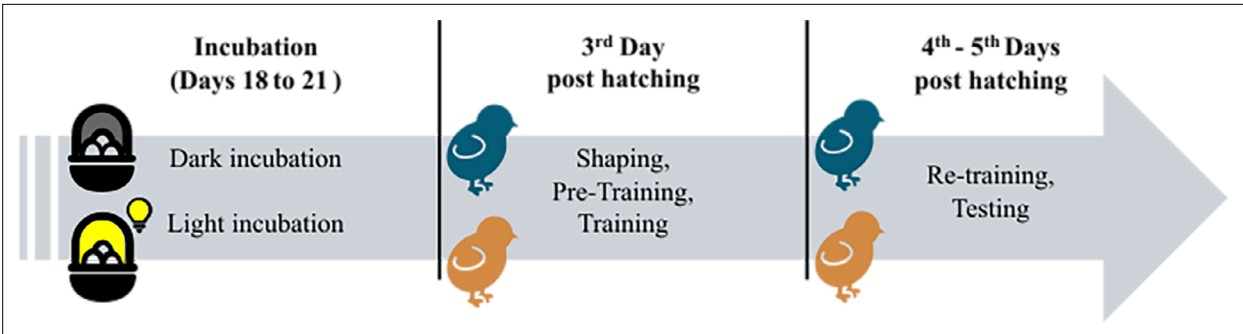

**Figure 4.** Time schedule of the experiments. During the last 3 days of incubation, eggs were incubated either in darkness (weakly lateralized chicks) or under light exposure (strongly lateralized chicks). When chicks were 3 days of age, the shaping procedure started, followed by pre-training and training. In the following days, chicks were re-trained and then underwent all four tests.

were rehoused in social groups with water and food ad libitum. On Friday afternoon, their fifth day of life, chicks were donated to local farmers (see *Figure 4* for a graphical illustration of the experimental schedule).

## Apparatus

The experimental apparatus was located in the experimental room, near the rearing room, and maintained at constant temperature and humidity (28°C; 70%). The apparatus consisted of a square arena constructed from green polypropylene (100×100×40 cm³) with the floor covered with wood shavings (*Figure 1*). Inside the arena were 10 identical red plastic caps (3 cm in diameter, 0.9 cm in height), each filled with wood shavings. To minimize the potential use of external cues, the entire setup was elevated and rotated randomly between trials or sessions (*Rugani et al., 2007*; *Rugani et al., 2010*). Bottle caps were frequently shuffled to ensure that the choices made by the chicks did not depend on some unique characteristics of the caps (*Rugani et al., 2016*; *Rugani et al., 2011*; *Rugani and Regolin, 2021*). The apparatus comprised two mirrored starting positions (15×15×10 cm³ boxes) outside opposing walls, one of which serves as a starting position (labeled as 'S.P.' in *Figure 1*). The starting boxes were designed to allow consistent visual input of the inner apparatus to the chicks. Access to the arena was provided through an entrance door made of green polypropylene (10×17 cm²), which could be lifted by a nylon thread.

## Shaping

In both experiments, the shaping, pre-training, and training procedures were conducted in the same setting. Shaping started on Wednesday morning after 2 hr of food deprivation (8:00–10:00 a.m.). This was essential in motivating foraging behavior during training and testing. During shaping, the array made of 10 items was centrally aligned along the median sagittal line, thus sagittally oriented, with respect to either starting point (*Figure 1A*), with the first cap positioned 28.5 cm from each entrance door. Each cap was positioned 1.44 cm apart from the subsequent one, with the overall array length being 43 cm. The array was situated 48.5 cm from either side wall.

The experimenter first introduced the chick into the arena for habituation, which lasted for about 2 min, allowing the bird to explore until it showed no signs of distress. Subsequently, the experimenter placed the chick into the starting point, and the shaping began. A piece of mealworm was placed and remained visible on the fourth cap to reinforce pecking behavior at that ordinal position. After the chick entered the arena and first pecked at any item, the trial was over, and the chick was immediately placed back into the starting box. If the chick did not peck at any item within 30 s, the experimenter used a metal stick to direct the chick to the fourth cap. After the chick had successfully pecked the fourth cap in 10 trials (whether consecutive or not), in the subsequent trials, the food was gradually covered with wood shavings until it was completely hidden. The shaping lasted for 10–15 min, followed by 30–40 min of rest back in the rearing cage with access to water but not to food.

## Pre-training

After the resting period, the pre-training began. From the pre-training, all items looked identical as the food reward in the correct (fourth) cap was completely buried in wood shavings. The chick had to complete three consecutive correct trials to reach the learning criterion and pass the pre-training phase (*Rugani et al., 2011*; *Rugani et al., 2016*; *Rugani and Regolin, 2020*; *Rugani and Regolin, 2021*). This usually took about 5–10 min. If the chick did not reach the learning criterion, the pre-training was repeated after 30 min of rest. If the chick again did not reach the criterion, it was excluded from the study.

## Training

Immediately after completion of the pre-training, chicks underwent training, comprising 20 trials, using the identical sequence employed in the pre-training phase. In each trial, only one choice was allowed, and the trial was terminated as soon as the chick pecked any item, with its choice being recorded. If the chick did not peck any item in 180 s, the trial was considered null and terminated. A choice was considered correct if the chick pecked at the 4th item. The learning criterion for passing the training phase was 8 correct trials out of 20 trials (*Rugani et al., 2011*; *Rugani et al., 2016*; *Rugani and Regolin, 2020*; *Rugani and Regolin, 2021*). If a subject did not achieve the learning criterion, after 40 min of rest, it underwent another pre-training and, if this was successful, the training began. Each chick had three chances to pass the training criterion. During this phase, five chicks were excluded in experiment 1 and nine in experiment 2 due to motivational or health problems, an exclusion rate that aligns with those of previous studies involving similar procedures (*Rugani et al., 2020a*; *Rugani et al., 2015b*). These subjects were subsequently replaced to maintain the predetermined sample size.

## Re-training

Re-training was conducted prior to every test. The re-training procedure was the same as pre-training and ended with three consecutive correct trials. Test sessions started immediately after re-training was completed.

## Test sessions

All chicks participated in all four tests: first the sagittal test, then the fronto-parallel binocular test. Thereafter, monocular tests were administered in counterbalanced order. The fronto-parallel monocular right test was conducted prior to the fronto-parallel monocular left test for *n*=24 chicks in experiment 1 and *n*=26 in experiment 2. The remaining chicks (*n*=24 in experiment 1, *n*=26 in experiment 2) underwent the fronto-parallel monocular tests in the reversed order. Experimenters and scorers were blinded to study aims during data collection and analysis.

## Sagittal test

The procedure for the sagittal test (*Figure 1A*) was the same as the training. The sagittal test consisted of 20 trials, and the time limit for each trial was 60 s. During testing, food reinforcement was available only in pre-established trials to prevent the extinction of responses over multiple unrewarded test trials (reinforced trials: 4, 5, 7, 10, 13, 14, 16, and 19; *Rugani et al., 2016*; *Rugani et al., 2011*; *Rugani and Regolin, 2021*; *Rugani and Regolin, 2020*). Thereafter, subjects rested for at least 60 min before entering the fronto-parallel tests.

In experiment 1, ordinal and spatial cues were available to identify the 4th correct item; in fact, the array was arranged as during training, with the length of the series being kept constant throughout the trials.

In experiment 2, to eliminate spatial cues to locate the 4th item, the inter-item distance varied between test trials (1.44 cm, 2.55 cm, 3.11 cm, and 3.66 cm), while remaining equally spaced within each trial, resulting in total array lengths of 43.0, 53.0, 58.0, and 63.0 cm, respectively. The first cap was set at 28.5 cm from the starting position.

## Fronto-parallel tests

In both experiments, the fronto-parallel test was conducted on each subject in three different conditions of vision (*Figure 1*). The binocular fronto-parallel test was always administered first. Then half

of the chicks underwent the left monocular fronto-parallel test and finally the right monocular fronto-parallel test, while the other half underwent the monocular tests in reverse order.

In the monocular fronto-parallel test, a temporary eye patch was carefully applied to restrict visual input to one of the chicks' eyes. The patch, made of removable paper tape, did not obstruct eyelid movements and enabled smooth removal post-testing without harming the subject. Before the actual test, chicks were habituated to wear the eye patch for about 15 min, during which they were closely monitored; any signs of distress or excessive scratching prompted immediate intervention to ensure the animals' well-being (*Rogers and Vallortigara, 2017*; *Rugani et al., 2016*; *Rugani and Regolin, 2021*).

In the three fronto-parallel tests, the array was rotated 90°, fronto-parallel with respect to the starting point (*Figure 1B*). Inter-item distances matched those in the sagittal test for each experiment. In this rotated array, both the 4th item from left (4L) and right (4R) were considered correct and rewarded during the pre-established trials (as described for the sagittal test). Each fronto-parallel test comprised 20 trials, with 2 hr rest periods between tests.

## Statistical analyses

In each trial, chicks were allowed a single peck. We recorded the selected item to calculate the percentage of responses at each position as [(number of pecks to a given item ÷ total number valid trials)×100] and averaged them separately for each group and test. We employed both frequentist and Bayesian statistical approaches, conducting corresponding Bayesian analyses for each frequentist test. We analyzed the group percentage for choosing each item above chance (10%), using Wilcoxon one-sample signed-rank tests with Bonferroni correction for multiple comparisons (data and significant results are reported in *Supplementary file 1*; *Supplementary file 2*) and one-sample Bayesian t-tests. To assess side bias in the fronto-parallel tests, we compared correct choices on the left (4L) vs. the right (4R) using paired t-tests, with Cohen's d as the effect size and Bonferroni as the correction method; moreover, we conducted two-sample Bayesian t-tests. Additionally, we tested whether brain lateralization influenced accuracy by comparing the percentage of correct choices (i.e. the selection of the 4th item in the sagittal test and of the 4L or 4R items in the fronto-parallel tests) between Li-chicks and Di-chicks using two-sample t-tests, with Cohen's d as the effect size and Bonferroni as the correction method; additionally, we conducted two-sample Bayesian t-tests.

Bayesian factors were computed using the BayesFactor package (*Morey and Rouder, 2012*). The analyses were conducted using R (version 4.3.1; R Core Team, 2022). We used the classification by *Lee and Wagenmakers, 2014*, to interpret BFs.

## Acknowledgements

European Union Horizon 2020 Research and Innovation program under the Marie Sklodowska-Curie Grant/Award Number: 795242 to RR; PRIN 2022, grant/award number: 202254RHRT to RR; PRIN 2022 PNRR, grant/award number: P2022TKY7B, to RR and LR; PRIN 2017: grant/award number: 2017PSRHPZ_003, to LR and RR.

## Additional information

### Funding

| Funder | Grant reference number | Author |
| --- | --- | --- |
| Marie Sklodowska-Curie Actions | 10.3030/795242 | Rosa Rugani |
| Ministero dell'Università e della Ricerca | PRIN 2022 202254RHRT | Rosa Rugani |
| Ministero dell'Università e della Ricerca | PRIN-PNRR 2022 P2022TKY7B | Lucia Regolin |
| Ministero dell'Università e della Ricerca | PRIN 2017 2017PSRHPZ | Lucia Regolin |

| Funder | Grant reference number | Author |
| --- | --- | --- |

The funders had no role in study design, data collection and interpretation, or the decision to submit the work for publication.

## Author contributions

Rosa Rugani, Conceptualization, Resources, Supervision, Funding acquisition, Validation, Investigation, Methodology, Writing – original draft, Project administration, Writing – review and editing; Matteo Macchinizzi, Yujia Zhang, Data curation, Formal analysis, Investigation, Writing – review and editing; Lucia Regolin, Supervision, Visualization, Writing – review and editing

## Author ORCIDs

Rosa Rugani ⓘ https://orcid.org/0000-0001-5294-6306
Matteo Macchinizzi ⓘ https://orcid.org/0009-0004-1203-4117
Yujia Zhang ⓘ https://orcid.org/0000-0002-0740-6383
Lucia Regolin ⓘ https://orcid.org/0000-0001-8960-0309

## Ethics

All experimental procedures employed were evaluated, approved, and conducted in strict adherence to the guidelines provided by the Ethical Committee of the University of Padova for Animal Experimentation (Organismo preposto al Benessere Animale, OPBA) and the Ministry of Health of the Italian Republic (Prot. N.9245, 17/01/2019). This comprehensive compliance addressed both national and European directives concerning animal research.

Reviewer #1 (Public review): https://doi.org/10.7554/eLife.106356.3.sa1
Reviewer #2 (Public review): https://doi.org/10.7554/eLife.106356.3.sa2
Author response https://doi.org/10.7554/eLife.106356.3.sa3

---

# Additional files

## Supplementary files

Supplementary file 1. Descriptive statistic—experiment 1. Descriptive statistics of the accuracy in selecting each of the 10 items in the sagittal test, and in selecting the 5 items on the left (L) and the 5 items on the right (R) in the fronto-parallel (FP) tests of experiment 1.

Supplementary file 2. Descriptive statistic—experiment 2. Descriptive statistics of the accuracy in selecting each of the 10 items in the sagittal test, and in selecting the 5 items on the left (L) and the 5 items on the right (R) in the fronto-parallel (FP) tests of experiment 2.

MDAR checklist

## Data availability

All data are available in the main text or the supplementary materials. Additional material containing metadata, row data, script used for the analysis, output of the script, and a folder with the same materials available for macOS for this article is available on Research Data Unipd.

The following dataset was generated:

| Author(s) | Year | Dataset title | Dataset URL | Database and Identifier |
| --- | --- | --- | --- | --- |
| Rosa R, Lucia R, Matteo M, Yujia Z | 2024 | Hatching with Numbers: How Pre-natal Experience Affects Chicks' Left-to-Right Mental Number Line | https://doi.org/10.25430/researchdata.cab.unipd.it.00001424 | Research Data UNIPD, 10.25430/researchdata.cab.unipd.it.00001424 |

---

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
