## [Editor Report · eLife Assessment]

This **fundamental** study demonstrates how a left-right bias in the relationship between numerical magnitude and space depends on brain lateralization. The evidence is **compelling** and will be of interest to researchers studying numerical cognition, brain lateralization, and cognitive brain development more broadly.

---

## [Referee Report · Reviewer #1 (Public review)]

Functional lateralization between the right and left hemispheres is reported widely in animal taxa, including humans. However, it remains largely speculative as to whether the lateralized brains have a cognitive gain or a sort of fitness advantage. In the present study, by making use of the advantages of domestic chicks as a model, the authors are successful in revealing that the lateralized brain is advantageous in the number sense, in which numerosity is associated with spatial arrangements of items. Behavioral evidence is strong enough to support their arguments. Brain lateralization was manipulated by light exposure during the terminal phase of incubation, and the left-to-right numerical representation appeared when the distance between items gave a reliable spatial cue. The light-exposure induced lateralization, though quite unique in avian species, together with the lack of intense inter-hemispheric direct connections (such as the corpus callosum in the mammalian cerebrum), was critical for the successful analysis in this study. Specification of the responsible neural substrates in the presumed right hemisphere is expected in future research. Comparable experimental manipulation in the mammalian brain must be developed to address this general question (functional significance of brain laterality) is also expected.

---

## [Referee Report · Reviewer #2 (Public review)]

Summary:

This is the first study to show how a L-R bias in the relationship between numerical magnitude and space depends on brain lateralisation, and moreover, how this is modulated by in ovo conditions.

Strengths:

Novel methodology for investigating the innateness and neural basis of a L-R bias in the relationship between number and space.

Weaknesses:

I would query the way the experiment was contextualised. They ask whether culture or innate pre-wiring determines the 'left-to-right orientation of the MNL [mental number line]'.

The term, 'Mental Number Line' is an inference from experimental tasks. One of the first experimental demonstrations of a preference or bias for small numbers in the left of space and larger numbers in the right of space, was more carefully described as the spatial-numerical association of response codes - the SNARC effect (Dehaene, S., Bossini, S., & Giraux, P. (1993). The mental representation of parity and numerical magnitude. Journal of Experimental Psychology: General, 122, 371-396).

This has meant that the background to the study is confusing. First, they note correctly that many other creatures, including insects can show this bias, though in none of these has neural lateralisation been shown to be a cause. Second, their clever experiment shows that an experimental manipulation creates the bias. If it were innate and common to other species, the experimental manipulation shouldn't matter. There would always be a L-R bias. Third, they seem to be asserting that humans have a left-to-right (L-R) MNL. This is highly contentious, and in some studies, reading direction affects it, as the original study by Dehaene et al showed; and in others, task affects direction (e.g. Bachtold, D., Baumüller, M., & Brugger, P. 1998). Stimulus-response compatibility in representational space. Neuropsychologia, 36, 731-735, not cited). Moreover, a very careful study of adult humans, found no L-R bias (Karolis, V., Iuculano, T., & Butterworth, B. (2011), not cited). Mapping numerical magnitudes along the right lines: Differentiating between scale and bias. Journal of Experimental Psychology: General, 140(4), 693-706. Indeed, Rugani et al claim, incorrectly, that the L-R bias was first reported by Galton in 1880. There are two errors here: first, Galton was reporting what he called 'visualised numerals' and are typically referred to now as 'number forms' - spontaneous and habitual conscious visual representations - not an inference from a number line task. Second, Galton reported right-to-left, circular, and vertical visualised numerals, and no simple left-to-right examples (Galton, F. (1880). Visualised numerals. Nature, 21, 252-256. So in fact did Bertillon, J. (1880). De la vision des nombres. La Nature, 378, 196-198, and more recently Seron, X., Pesenti, M., Noël, M.-P., Deloche, G., & Cornet, J.-A. (1992). Images of numbers, or "When 98 is upper left and 6 sky blue". Cognition, 44, 159-196, and Tang, J., Ward, J., & Butterworth, B. (2008). Number forms in the brain. Journal of Cognitive Neuroscience, 20(9), 1547-1556.

If the authors are committed to chicks' MN Line they should test a series of numbers showing that the bias to left is greater for 2 and 3 than for 4 etc.

What does all this mean? I think that the experiment should absolutely be published in eLife, but the paper should be shorn of its misleading contextualisation, including the term 'Mental Number Line'. The authors also speculate, usefully, on why chicks and other species might have a L-R bias. I don't think the speculations are convincing, but at least if there is an evolutionary basis for the bias, it should at least be discussed.

In fact, I think it would make a very interesting special issue to bring up to date how and why the L-R bias exists, and where and why it does not.

Karolis, V., Iuculano, T., & Butterworth, B. (2011). Mapping numerical magnitudes along the right lines: Differentiating between scale and bias. Journal of Experimental Psychology: General, 140(4), 693-706. doi:10.1037/a0024255

Review of the revised version:

The background and terminology in the text have been significantly altered and clarified: Spatial Numerical Association (SNA) instead of Mental Number Line (MNL) in the text, but with a discussion about how SNA might be the basis of MNL. This entails a link from SNA - a bias - to mental representation of a sequence of numerical magnitudes, which will need to be spelt out in subsequent work with a sequence of numbers rather than a single number, in this case 4. Could the effect be generalised to much larger numbers?

Although the relationship between number and space seems fundamental, the key question is why the L-R SNA bias should exist at all. The authors take on this challenge and make important arguments for the evolutionary advantage of the bias is (see lines 138ff, 375ff, 444ff), though this is likely still to be controversial.

Subsequent work may clarify its interaction of brain lateralisation with culture, notably reading and writing direction (e.g. Dehaene, S., Bossini, S., & Giraux, P. (1993). The mental representation of parity and numerical magnitude. Journal of Experimental Psychology: General, 122, 371-396), though this relationship has exceptions and challenges (e.g. Karolis, V., Iuculano, T., & Butterworth, B. (2011). Mapping numerical magnitudes along the right lines: Differentiating between scale and bias. Journal of Experimental Psychology: General, 140(4), 693-706).

For example, would humans with more lateralised brains show a stronger bias? Would humans with reverse lateralisation show a R-L SNA?

---

## [Author Response]

The following is the authors’ response to the original reviews.

**Reviewer #1 (Public review):**
Functional lateralization between the right and left hemispheres is reported widely in animal taxa, including humans. However, it remains largely speculative as to whether the lateralized brains have a cognitive gain or a sort of fitness advantage. In the present study, by making use of the advantages of domestic chicks as a model, the authors are successful in revealing that the lateralized brain is advantageous in the number sense, in which numerosity is associated with spatial arrangements of items. Behavioral evidence is strong enough to support their arguments. Brain lateralization was manipulated by light exposure during the terminal phase of incubation, and the left-to-right numerical representation appeared when the distance between items gave a reliable spatial cue. The light-exposure induced lateralization, though quite unique in avian species, together with the lack of intense inter-hemispheric direct connections (such as the corpus callosum in the mammalian cerebrum), was critical for the successful analysis in this study. Specification of the responsible neural substrates in the presumed right hemisphere is expected in future research. Comparable experimental manipulation in the mammalian brain must be developed to address this general question (functional significance of brain laterality) is also expected.

We sincerely appreciate the Reviewer's insightful feedback and his/her recognition of the key contributions of our study.

**Reviewer #2 (Public review):**
Summary:This is the first study to show how a L-R bias in the relationship between numerical magnitude and space depends on brain lateralisation, and moreover, how is modulated by in ovo conditions.Strengths:Novel methodology for investigating the innateness and neural basis of an L-R bias in the relationship between number and space.

We would like to thank the Reviewer for their valuable feedback and for highlighting the key contributions of our study.

Weaknesses:I would query the way the experiment was contextualised. They ask whether culture or innate pre-wiring determines the 'left-to-right orientation of the MNL [mental number line]'.

We thank the Reviewer for raising this point, which has allowed us to provide a more detailed explanation of this aspect. Rather than framing the left-to-right orientation of the mental number line (MNL) as exclusively determined by either cultural influences or innate pre-wiring, our study highlights the role of environmental stimulation. Specifically, prenatal light exposure can shape hemispheric specialization, which in turn contributes to spatial biases in numerical processing. Please see lines 115-118.

The term, 'Mental Number Line' is an inference from experimental tasks. One of the first experimental demonstrations of a preference or bias for small numbers in the left of space and larger numbers in the right of space, was more carefully described as the spatial-numerical association of response codes - the SNARC effect (Dehaene, S., Bossini, S., & Giraux, P. (1993). The mental representation of parity and numerical magnitude. Journal of Experimental Psychology: General, 122, 371-396).

We have refined our description of the MNL and SNARC effect to ensure conceptual accuracy in the revised manuscript; please see lines 53-59.

This has meant that the background to the study is confusing. First, the authors note, correctly, that many other creatures, including insects, can show this bias, though in none of these has neural lateralisation been shown to be a cause. Second, their clever experiment shows that an experimental manipulation creates the bias. If it were innate and common to other species, the experimental manipulation shouldn't matter. There would always be an L-R bias. Third, they seem to be asserting that humans have a left-to-right (L-R) MNL. This is highly contentious, and in some studies, reading direction affects it, as the original study by Dehaene et al showed; and in others, task affects direction (e.g. Bachtold, D., Baumüller, M., & Brugger, P. (1998). Stimulus-response compatibility in representational space. Neuropsychologia, 36, 731-735, not cited). Moreover, a very careful study of adult humans, found no L-R bias (Karolis, V., Iuculano, T., & Butterworth, B. (2011), not cited, Mapping numerical magnitudes along the right lines: Differentiating between scale and bias. Journal of Experimental Psychology: General, 140(4), 693-706). Indeed, Rugani et al claim, incorrectly, that the L-R bias was first reported by Galton in 1880. There are two errors here: first, Galton was reporting what he called 'visualised numerals', which are typically referred to now as 'number forms' - spontaneous and habitual conscious visual representations - not an inference from a number line task. Second, Galton reported right-to-left, circular, and vertical visualised numerals, and no simple left-to-right examples (Galton, F. (1880). Visualised numerals. Nature, 21, 252-256.). So in fact did Bertillon, J. (1880). De la vision des nombres. La Nature, 378, 196-198, and more recently Seron, X., Pesenti, M., Noël, M.-P., Deloche, G., & Cornet, J.-A. (1992). Images of numbers, or "When 98 is upper left and 6 sky blue". Cognition, 44, 159-196, and Tang, J., Ward, J., & Butterworth, B. (2008). Number forms in the brain. Journal of Cognitive Neuroscience, 20(9), 1547-1556.

We sincerely appreciate the opportunity to discuss numerical spatialization in greater detail. We have clarified that an innate predisposition to spatialize numerosity does not necessarily exclude the influence of environmental stimulation and experience. We have proposed an integrative perspective, incorporating both cultural and innate factors, suggesting that numerical spatialization originates from neural foundations while remaining flexible and modifiable by experience and contextual influences. Please see lines 69–75.

We have incorporated the Reviewer’s suggestions and cited all the recommended papers; please see lines 47–75.

If the authors are committed to chicks' MN Line they should test a series of numbers showing that the bias to the left is greater for 2 and 3 than for 4, etc.What does all this mean? I think that the paper should be shorn of its misleading contextualisation, including the term 'Mental Number Line'. The authors also speculate, usefully, on why chicks and other species might have a L-R bias. I don't think the speculations are convincing, but at least if there is an evolutionary basis for the bias, it should at least be discussed.

In the revised version of the manuscript, we have resorted to adopt the Spatial Numerical Association (SNA). We thank the Reviewer for this valuable comment.

We appreciated the Reviewer’s suggestion regarding the evolutionary basis of lateralization and have included considerations of its relevance in chicks and other species; please see lines 143-151 and 381-386.

This paper is very interesting with its focus on why the L-R bias exists, and where and why it does not.

We wish to thank the Reviewer again for his/her work.

**Reviewer #1(Public review)**
(1) Introduction needs to be edited to make it much more concise and shorter. Hypotheses (from line 67 to 81) and predictions (from line 107 to 124) must be thoroughly rephrased, because (a) general readers are not familiar with the hypotheses (emotional valence and BAFT), (b) the hypotheses may or may not be mutually exclusive, and therefore (c) the logical linkage between the hypotheses and the predicted results are not necessarily clear. Most general readers may be embarrassed by the apparently complicated logical constructs of this study. Instead, it is recommended that focal spotlight should be given to the issue of functional contributions of brain lateralization to the cognitive development of number sense.

We thank the Reviewer for these comments, which allowed us to improve the clarity of our hypotheses and predictions. We thoroughly rephrased them to ensure they are accessible to general readers and specified that the models may or may not be mutually exclusive. Additionally, we highlighted the functional contributions of brain lateralization to the cognitive development of number sense, addressing the suggested focal point. While we have shortened the introduction, we opted to retain essential background information to ensure readers are well-informed about the relevant scientific literature. Please review the entire introduction, particularly lines 84–118 and 218.

(2) In relation to the above (a), abbreviations need to be reexamined. MNL (mental number line) appears early on lines 27 and 49, whereas the possibly related conceptual term SNA appeared first on line 213, without specification to "spatial numerical association".

We thank the Reviewer for bringing this to our attention. We have addressed the suggestions, and the term SNA has been used specifically to refer to numerical spatialization in non-human animals. Please see lines 27-30.

(3) By the way, what difference is there between MNL and SNA? Please specify the difference if it is important. If not important, is it possible that one of these two is consistently used in this report, at least in the Introduction?

We clarified the distinction between MNL and SNA and have consistently used SNA in this report; please see lines 47-75.

(4) In relation to the above (a and b), clarification of the hypotheses and their abbreviations in the form of a table or a graphical representation will strongly reinforce the general readers' understanding. It is also possible that some of these hypotheses are discussed later in the Discussion, rather than in Introduction.

We appreciated this suggestion and have now clarified the hypotheses, also providing a table/graphical representation, aiming to enhance accessibility for general readers; please see lines 110-118, and 218.

(5) Figures 1 and 2 are transparent and easily understandable; however, the statistical details in the Results may bother the readers as the main points are doubly represented in Figures 1, 2, and Table 1. These (statistics and Table 1) may go to the supplementary file, if the editor agrees.

We would prefer to keep Table 1 and the statistical details as part of the main article to provide readers with a comprehensive overview of the experimental results. However, if the editors also suggest to move them to the supplementary file, we are open to making this adjustment.

(6) In Figure 1D and E, and text lines 139-140. Figure 1D shows that the chick is looking monocularly by the right eye, but the text (line 139) says "left eye in use. Is it correct?

We thank the reviewer for pointing out this incongruity. We have corrected the text to align with Figure 1D and E; please see lines 180-181.

(7) Methods. The behavioral experiment was initiated on Wednesday (8 a.m.; line 479), but at what age? At what post-hatch day was the experiment terminated? A simple graphical illustration of the schedule will be quite helpful.

We have added the requested details, specifying that experiments began on the third post-hatch day and ended on the fifth day; please see lines 533-539.

Additionally, we have included a graphical illustration of the schedule to enhance clarity; please see line 666.

(8) Methods. How many chicks were excluded from the study in the course of Pre-training (line 525) and Training (line 535-536)? Was the exclusion rate high, or just negligible?

We appreciate the reviewer's suggestion. We have now included the number of subjects excluded during the training phase; please see lines 593-597.

We wish to thank the Reviewer again for his/her work.